# The Relationship of the Atlantic Diet with Cardiovascular Risk Factors and Markers of Arterial Stiffness in Adults without Cardiovascular Disease

**DOI:** 10.3390/nu11040742

**Published:** 2019-03-29

**Authors:** Carmela Rodríguez-Martín, Luis Garcia-Ortiz, Emiliano Rodriguez-Sanchez, Carlos Martin-Cantera, Alba Soriano-Cano, Maria S. Arietaleanizbeaskoa, Jose F. Magdalena-Belio, Marta Menendez-Suarez, Jose A. Maderuelo-Fernandez, Cristina Lugones-Sanchez, Manuel A. Gómez-Marcos, José I. Recio-Rodríguez

**Affiliations:** 1Institute of Biomedical Research of Salamanca (IBSAL), Primary Health Care Research Unit, La Alamedilla Health Center. Health Service of Castilla y León (SACYL), Primary Care Prevention and Health Promotion Research Network (REDIAPP), 37003 Salamanca, Spain; carmelarroma@hotmail.com (C.R.-M.); lgarciao@usal.es (L.G.-O.); emiliano@usal.es (E.R.-S.); jmaderuelo@saludcastillayleon.es (J.A.M.-F.); crislugsa@gmail.com (C.L.-S.); magomez@usal.es (M.A.G.-M.); 2Department of Biomedical and Diagnostic Sciences, University of Salamanca, 37008 Salamanca, Spain; 3Primary Health Care Research Unit of Barcelona, Institute IDIAP Jordi Gol, 08007 Barcelona, Spain; 18986cmc@comb.cat; 4Universidad de Castilla-La Mancha, Health and Social Research Center, 16071 Cuenca, Spain; alba.soriano@uclm.es; 5Primary Health Care Research Unit of Bizkaia. Basque Health Service-Osakidetza, 48014 Bilbao, Spain; MARIASOLEDAD.ARIETALEANIZBEASKOASARABIA@osakidetza.eus; 6Torre Ramona Health Center, Aragon Health Service, Instituto de Investigacion Sanitaria IIS-Aragón, 50013 Zaragoza, Spain; jfmagdalenab@gmail.com; 7Casa del Barco Health Center, Castilla and León Health Service, 47007 Valladolid, Spain; martamenendezsuarez@gmail.com; 8Department of Medicine, University of Salamanca, 37008 Salamanca, Spain; 9Department of Nursing and Physiotherapy, University of Salamanca, 37008 Salamanca, Spain

**Keywords:** healthy diet, diet, Mediterranean, vascular stiffness, obesity abdominal, risk factors

## Abstract

Background: Studying the adherence of the population to the Atlantic Diet (AD) could be simplified by an easy and quickly applied dietary index. The aim of this study is to analyse the relationship of an index measuring compliance with recommendations regarding the Atlantic diet and physical activity with cardiovascular disease risk factors, cardiovascular risk factors, obesity indexes and arterial stiffness markers. Methods: We included 791 individuals from the EVIDENT study (lifestyles and arterial ageing), (52.3 ± 12 years, 61.7% women) without cardiovascular disease. Compliance with recommendations on AD was collected through the responses to a food frequency questionnaire, while physical activity was measured by accelerometer. The number of recommendations being met was estimated using a global scale between 0 and 14 points (a higher score representing greater adherence). Blood pressure, plasma lipid and glucose values and obesity rates were measured. Cardiovascular risk was estimated with the Framingham equation. Results: In the overall sample, 184 individuals (23.3%) scored between 0–3 on the 14-point index we created, 308 (38.9%) between 4 and 5 points, and 299 (37.8%) 6 or more points. The results of multivariate analysis yield a common tendency in which the group with an adherence score of at least 6 points shows lower figures for total cholesterol (*p* = 0.007) and triglycerides (*p* = 0.002). Similarly, overall cardiovascular risk in this group is the lowest (*p* < 0.001), as is pulse wave velocity (*p* = 0.050) and the mean values of the obesity indexes studied (*p* < 0.05 in all cases). Conclusion: The rate of compliance with the Atlantic diet and physical activity shows that greater adherence to these recommendations is linked to lower cardiovascular risk, lower total cholesterol and triglycerides, lower rates of obesity and lower pulse wave velocity values.

## 1. Introduction

In recent decades, the study of the effects of nutrition on health has undergone a change in approach, with the focus moving from isolated nutrients to dietary patterns [1]. This change is partly due to the difficulty of studying the effects of individual nutrients since these are consumed as part of a dietary pattern alongside other nutrients, among which complex interactions take place [2,3,4]. The analysis of a single nutrient in this context may thus be affected by a confounding phenomenon [5] and generate erroneous associations [6]. Some studies complement the analysis of the effect of these dietary patterns on health by analysing the principal component and/or using a cluster analysis. These analyses have been able, for example, to determine that within the concept of a healthy eating pattern, the consumption of a diet rich in cereals, fish, fruit and vegetables is associated with a healthier metabolic profile [7]. They have also allow determining that skipping breakfast or the afternoon snack are risk factors for poor cardiovascular health [8].

Among dietary patterns, the most studied has been the Mediterranean diet and the dietary approaches to stop hypertension (DASH). The Mediterranean diet has shown a reduction in both the incidence and prevalence of chronic diseases like cardiovascular disease, cancer, metabolic syndrome, diabetes, neurodegenerative diseases, as well as a reduction in overall mortality [4]. This diet is considered one of the best dietary patterns in the framework of healthy lifestyle, probably due to the combination of many elements with antioxidant and anti-inflammatory properties [3], thereby constituting a useful tool for the prevention of cardiovascular disease [2,5,6] and making it one of the best analysed in relation to cardiovascular risk and other health outcomes [9]. The DASH dietary pattern, was originally developed to treat hypertension without medication and was associated with substantially lower risk of coronary artery disease and stroke mortality and other cardiometabolic advantages [10,11,12].

However, more recently, a pattern known as the Atlantic Diet (AD), representative of the traditional diet of Portugal and Galicia (Northwest Spain) has been focusing the attention. Although it has many elements in common with the Mediterranean diet (consumption of fruits and vegetables, nuts and olive oil) and with the DASH dietary pattern (fruits, vegetables nuts, legumes or moderate consumption of low fat dairy), the AD has some differentiating characteristics such as the increased intake of fish and seafood, potatoes, broths with meat and cabbage, and moderate consumption of lean meats [13]. To date, while available evidence regarding this dietary pattern is not particularly extensive, published scientific studies have shown that it can provide important health benefits. In a case-control study conducted in 2010 in Porto by Oliveira et al. [14], an inverse association between adherence to AD and a lower probability of non-fatal myocardial infarction was reported. Guallar-Castillon et al. [15], examined the association between Southern European AD (SEAD) and various biomarkers of coronary risk, blood pressure and anthropometry. Greater adherence to the SEAD was associated with lower levels of C-reactive protein in plasma, plasma triglycerides, insulin resistance index, albumin in urine, urine albumin-creatinine ratio, and systolic blood pressure.

Studying the adherence of the population to this dietary pattern and its relationship with cardiovascular disease, as well as possible comparisons with other dietary patterns, could be simplified by an easy and quickly applied dietary index. The aim of this study is to develop a dietary index, based on a food frequency questionnaire, which would allow a quick and easy assessment of compliance with the recommendations regarding the Atlantic diet and physical activity and an analysis of its association with cardiovascular disease risk score as primary outcome and cardiovascular risk factors, obesity indexes and arterial stiffness markers as secondary outcomes.

## 2. Materials and Methods

### 2.1. Study Design

The results of this study are a subanalysis of the EVIDENT 2 study (Lifestyles and arterial aging). This article presents the data from the baseline assessment alone. The EVIDENT 2 study protocol [16], with a description of data gathering methods, has been published previously. The EVIDENT 2 was a multi-centre clinical trial aiming to assess the effect of a smartphone application with brief counselling on improving the lifestyles in the general population.

### 2.2. Study Population

The EVIDENT 2 study participants were selected through random sampling from offices of general practitioners in six health centers from Spain. For this analysis, among the 833 participants in the EVIDENT 2 study, we included a total of 791 individuals. The remaining 42 participants did not have valid dietary records at the baseline assessment visit and collected using a food frequency questionnaire (FFQ). Other measures included in this study (cardiovascular risk factors and obesity indexes) were present on the 791 participants included. Exclusion criteria were as follows: being over 70 years of age, having cardiovascular disease, heart failure, moderate or severe chronic obstructive pulmonary disease, musculoskeletal disease which prevented walking, advanced liver, lung or kidney disease, severe mental illness, oncological disease treated and diagnosed in the 5 years prior to the beginning of the study, terminal illness, and pregnancy.

### 2.3. Ethics Approval and Consent to Participate

The study was approved by the clinical research ethics committee (CEIC) of the health care area of Salamanca (“CEIC of Area de Salud de Salamanca”, 21 June 2013) as a coordinating centre. It was also approved by the ethics committees of the five collaborating centres (“CEIC of Aragón (CEICA), CEIC of IDIAP Jordi Gol, CEIC of Euskadi (CEIC-E), CEIC of Castilla la Mancha and CEIC of the Area de Salud de Valladolid Oeste”). Subjects signed informed consent forms prior to inclusion in the study, in accordance with the Declaration of Helsinki. Trial registration: ClinicalTrials.gov Identifier: NCT02016014

### 2.4. Variables and Measuring Instruments

Assessment of compliance with recommendations on the Atlantic diet and physical activity: Dietary intake was collected by means of a semiquantitative, self-administered food frequency questionnaire (FFQ), using the previous year as the reference period. This questionnaire has been validated for energy, nutrient intake and food groups, against three-day dietary records in Spain [17] and includes 137 foods frequently used among the reference population. After receiving instructions from the study staff, participants indicated the frequency with which each food item was consumed over the last year using a Likert scale of nine options (never or almost never, 1–3 times a month, once a week, 2–4 times a week, 5–6 times a week, once a day, 2–3 times a day, 4–6 times a day or more than 6 times a day). Based on the results, we estimated the daily energy consumption (kcal), as well as the daily intake of macro and micronutrients (g). With the data of the FFQ, we have calculated an index of adherence to the Atlantic Diet, adapting the recommendations provided by Oliveira et al. [14], Calvo-Malvar et al. [18] and Vaz Velho et al. [13]. Oliveira et al. [14] developed an index based on a food frequency questionnaire. This index consists of nine food groups (fresh fish, dried salt cod, red meat and pork products, dairy products, legumes and vegetables, vegetable soup, potatoes, whole-grain bread, and wine). Subsequently, Calvo-Malvar et al. [18] established a series of recommendations for good compliance with this dietary pattern which, in addition to previous ones, included olive oil, fruit, eggs, nuts and other additional items to reduce the consumption of foods rich in animal fats, sweets or sugar-sweetened beverages. Finally, Vaz Velho [13], and Tojo et al. [19] included in the AD pyramid performing daily physical activity moderate to high intensity. All these recommendations were put together for the calculation of a 14-item index (Table 1). Compliance with each of the criteria defined in Table 1 scored one point, with the overall score ranging from 0 to 14 points (the higher the score, the greater the adherence). After the calculation of the index, the sample has been divided into 3 groups, using the integer values closest to the tertiles according to the number of AD recommendations met (0–3 points, 4–5 points and ≥6 points).

Assessment of habitual physical activity: Physical activity was measured using GT3X accelerometers, previously validated [20]. Data collected includes time spent daily (min/day) on moderate-vigorous activities. The device was worn for seven consecutive days.

Assessment of cardiovascular risk factors: For blood pressure, three readings were taken on each arm, using the mean of the last two from the arm with the highest values. Measurements were obtained with an oscillometric device, OMRON M10-IT (Omron Health Care, Kyoto, Japan), following the recommendations of the European Hypertension Society [21]. For the analysis of the laboratory variables, a blood sample was taken after a 12-h fast. The following values of plasma glucose, hemoglobin A1c (HbA1c) and lipid profile were measured: total plasma cholesterol, high density lipoprotein cholesterol (HDL-C) and triglycerides. Low density lipoprotein cholesterol (LDL-C) was estimated using the Friedewald equation except in subjects who had triglyceride levels ≥300 mg/dL (*n* = 10), in which case we used 299 mg/dL for the calculation of triglycerides. Hypertension was recorded with systolic blood pressure values of ≥140 mmHg and/or diastolic blood pressure values of ≥90 mmHg or if the subject was on antihypertensive medication [22]. Type 2 diabetes mellitus was present with HbA1c ≥ 6.5%, fasting plasma glucose ≥ 126 mg/dL, 2 h of plasma glucose ≥ 200 mg/dL during an oral glucose tolerance test, or random plasma glucose ≥ 200 mg/dl for a patient with classic symptoms of hyperglycemia or hyperglycaemic crisis or under antidiabetic treatment [23]. Information regarding drugs used for hypertension, type 2 diabetes and dyslipidemia was gathered by accessing data from the electronic medical records and then confirmed by the subjects themselves. Smoking history was assessed through questions about the participant’s smoking status (smoker/non-smoker). We considered smokers to be subjects who currently smoke or who stopped smoking less than one year ago.

Cardiovascular risk assessment (CVR): This was estimated using the published risk equation based on the Framingham study [24] to assess general cardiovascular disease risk and 10-year risk of individual cardiovascular events (coronary, cerebrovascular, and peripheral arterial disease and heart failure). Risk factors used include age, total cholesterol, high-density lipoprotein cholesterol and SBP as quantitative variables, and gender, drug treatment for hypertension, smoking and history of diabetes mellitus as dichotomous variables. The 791 participants who meet the selection criteria allow detecting a minimum difference in cardiovascular risk of three percentage points between two of the three groups into which we have classified AD adherence, assuming an alpha risk of 0.05 and a beta risk of 0.2 in bilateral contrast. This minimum difference was based on the results found after a diet and aerobic exercise program intervention, which achieved a reduction of 3.0 points on cardiovascular risk [25].

Anthropometric variables: Body weight was measured twice with an approved electronic scale (Seca 770, Medical Scale and Measurement Systems, Birmingham, UK) after calibration (accuracy ± 0.1 kg), with readings rounded to 100 g. Height was measured with a stadiometer (Seca 222, Medical Scale and Measurement Systems, Birmingham, UK), recording the average of two measurements. Body mass index (BMI) was calculated using the formula weight (kg) divided by height squared (m^2^). Waist and hip circumference were measured using a flexible tape measure following the recommendations of the Spanish Society for the Study of Obesity in 2007 [26]. All measurements were taken with the patient standing, barefoot and in light clothing. With these values, the waist-height index and the waist-hip index were subsequently calculated by means of the ratio of waist circumference to height or hip circumference, respectively. The adiposity index was based on the equation (Adiposity index = ((hip circumference)/((height) × (1.5)) − 18)) [27].

Analysis of pulse wave velocity (PWV): Pulse wave velocity was recorded in the Salamanca cohort (*n* = 291). This cohort had similar age and gender distribution to that the entire sample. The measurement was made with the patient in supine position using the SphymgoCor System (AtCor Medical Pty Ltd. Head Office, West Ryde, Australia), following the recommendations of the Van Bortel et al. consensus [28]. The pulse waves of the carotid and femoral arteries were analysed, and the delay was estimated with respect to the ECG wave. A tape measure was used to determine the distance from the sternal fork to the carotid and femoral artery sensor locations. Velocity was estimated in m/second.

### 2.5. Statistical Analysis

Results are expressed as means ± standard deviation for quantitative variables and by frequency distribution in the case of qualitative variables. The differences of means for continuous variables across the three groups were assessed through one-way analysis of variance (ANOVA) for independent samples, using the DMS method in post-hoc contrasts. An analysis of the covariance (ANCOVA) was also carried out, including as an independent variable the Atlantic diet and lifestyle index divided into the three categories described above, and as dependent variables cardiovascular risk, blood pressure, cholesterol, triglycerides, glucose, pulse wave velocity and anthropometric markers such as BMI, adiposity index, waist/height and waist/hip ratio. This analysis was adjusted only for the energy intake in the case of the cardiovascular risk to avoid collinearity. For the rest of the dependent variables, the analysis was controlled for age, gender and calorie intake (kcal) in all models, for the use of antihypertensive drugs in the case of blood pressure, for lipid-lowering drugs in the case of cholesterol and triglycerides, for antidiabetics in the case of glucose, and for antihypertensive drug treatment, blood pressure and smoking status in the pulse wave velocity analysis. Statistical analysis was performed with IBM SPSS Statistics for Windows, Version 23.0. (Armonk, NY, USA: IBM Corp.). A *p* value of <0.05 was considered statistically significant.

## 3. Results

Of the 791 individuals selected (52.3 ± 12 years, 61.7% women), 33.1% were hypertensive and 7.3% had type 2 diabetes mellitus (Table 2). The scores on the 14-point index scale of the total sample were as follows: 184 participants (23.3%) registered 0–3, 308 (38.9%) between 4 and 5 points, and 299 (37.8%) 6 or more points. The average index score assessing AD adherence was 4.9 points ± 1.7. Among the 14 items proposed, the one with the highest level of adherence (90.5%) was the consumption of at least three servings per week of fish or seafood. Conversely, the consumption of at least six servings/day of bread, cereals, wholegrain cereals, rice, pasta and potatoes was the item with the lowest compliance percentage (2.8%) (Table 3).

There are differences between the three groups with respect to energy consumption (kcal). The diet of the group with the highest adherence (6 points or more) has more calories, together with a lower percentage of saturated fat and a higher percentage of proteins, with the total amount of fat being similar to that consumed in the other groups (Table 4).

In terms of groups, the one comprising individuals with an AD adherence score of at least 6 points was characterized by being older (55.1 years) and having a higher percentage of women (68.9%) among them. In an unadjusted model, no differences were found between the groups with different AD adherence in either the variables related to cardiovascular risk factors or in the use of drugs. No differences were found in the figures for blood pressure or metabolic control, nor in the global values of cardiovascular risk or carotid-femoral pulse wave velocity. However, the group with the highest adherence showed lower levels of plasma triglycerides, as well as lower BMI and waist circumference values (Table 5).

In the adjusted model, controlling for potential confounding variables, the multivariate analysis yields a common tendency in which the group of those adhering to at least 6 points in the adherence index show lower total cholesterol figures (*p* = 0.007), and triglycerides (*p* = 0.002). Systolic blood pressure and glucose follow the same trend, without reaching statistical significance. Similarly, their overall cardiovascular risk is the lowest (*p* < 0.001), as is their pulse wave velocity figure (*p* = 0.050) (Table 5). Also, higher compliance with AD items is associated with lower figures of body mass index, waist circumference and waist-height index. 

## 4. Discussion

Developed from a food frequency questionnaire, the index created to assess the Atlantic Diet (AD) and lifestyle allows adherence to the recommendations described for this dietary pattern to be quantified. The main results show that greater compliance with these recommendations is associated with lower cardiovascular risk and lower figures for some cardiovascular risk factors such as cholesterol and triglycerides. Likewise, a higher score in this index is linked to lower figures in all the anthropometric markers studied. Finally, this study explores for the first time the relationship between this dietary pattern and arterial stiffness, showing an inverse relationship between them. The higher the level of adherence, the lower the pulse wave velocity.

The study of the health effects of certain dietary patterns requires valid instruments which are easy to administer in clinical practice. The PREDIMED study group (Effects of the Mediterranean diet in the primary prevention of cardiovascular disease) developed a questionnaire of 14 items [29] to assess the adherence to the Mediterranean Diet which has been used in innumerable subsequent studies. This questionnaire correlated very strongly with a food frequency questionnaire [29]. However, although AD has also been observed to have significant positive effects on health, there is no widely used questionnaire to assess compliance with its recommendations. The index created by our group, presented in this manuscript, includes all these recommendations, adapting the responses to a food frequency questionnaire widely used in other research [30,31]. The composition and validation of an AD adherence questionnaire using this index may be of interest in initiating a debate in which both dietary patterns (Mediterranean and Atlantic) can be compared in terms of predicting cardiovascular disease.

To date, not a great deal of evidence has been generated regarding the relationship of the AD dietary pattern with health. The work of Guallar-Castillon et al. [15] has yielded the most outstanding results. This study used the nine-component index, which we described in the previous paragraph, finding an average compliance of 2.9 points out of 9 possible pints among the 10,231 individuals studied. This represents relatively low adherence to AD and coincides with that found in the EVIDENT study (4.9 points ± 1.7) out of 14 possible points. Both studies concur in that greater compliance with the recommendations of this dietary pattern is related to lower values for blood pressure or plasma levels of cholesterol and triglycerides. However, the EVIDENT study is the first to explore the possible association with global cardiovascular risk, calculated by a risk equation. Among the possible explanations for the results found in this study, we can point out that individually, many of the components of AD have been associated with cardiovascular disease, including the clearest distinguishing aspect with respect to the Mediterranean diet or the DASH diet, which is a high consumption of fish [32]. The creation of this index allows measuring the combined effect of all these components, as well as their possible interactions, on cardiovascular risk or cardiovascular risk factors. Oliveira et al. [14] concluded that adherence to AD was associated with a lower probability of non-fatal myocardial infarction. These results could potentially be explained by the possible relationship indicated in this study, considering carotid-femoral pulse wave velocity, a marker of arterial stiffness, as a surrogate marker in the development of cardiovascular disease [33]. These results may therefore be the beginning of a line of research on the effects of AD in subclinical atherosclerosis assessed through arterial stiffness. Furthermore, the association of the AD index with the anthropometric markers exploring both general obesity (BMI, adiposity index) and abdominal obesity (waist-hip index and waist-height index) is also of relevance. This relationship may be mediated by the increase in some inflammatory markers such as insulin resistance or C-reactive protein [15]. The GALIAT clinical trial [18] will clarify these issues as it is the first clinical trial to analyse the effects of AD on adiposity markers.

The present study presents a series of limitations. The design of the study (being cross-sectional) does not allow causal relationships to be established between compliance with the recommendations of the AD and the variables studied, but it may be useful when proposing intervention or longitudinal studies to clarify the role of this dietary profile on health like those developed for the DASH diet or the Mediterranean diet. The AD index was created using the responses of the participants to a food frequency questionnaire which was previously validated in the reference population (Spanish population). Some studies indicate that FFQ may overestimate food consumption compared to other assessment instruments [34,35,36]. The population used for the analysis presented comprised participants of a clinical trial (EVIDENT 2), the aim of which was to measure the effectiveness of using a smartphone application in the improvement of lifestyles. However, the data used from the food frequency questionnaire are those corresponding to the first baseline assessment visit, at which stage no intervention of any kind had yet been carried out.

## 5. Conclusions

Based on the rate of compliance with the Atlantic diet and the lifestyle, developed from a food frequency questionnaire, the main results show that greater compliance with these recommendations is related to lower cardiovascular risk and lower average values for cholesterol and triglycerides, lower rates of obesity and lower figures for pulse wave velocity.

## Figures and Tables

**Table 1 nutrients-11-00742-t001:** Composition of the ATLANTIC diet and lifestyle index.

Components	Frequency	Servings Equivalents
Bread, cereals, wholegrain cereals, rice, pasta and potatoes	≥6 servings/day	1 serving: 75 g bread, 30 g cereals, 60 g rice, 60 g pasta, 150 g potatoes
Olive oil	≥3 servings/day	1 serving: 1 spoon
Fresh fruit	≥3 servings/day	1 serving: 1 piece or serving
Vegetables	≥2 servings/day	1 serving: 200 g
Dairy products	≥3 servings/day	1 serving: 200 g milk, 155 g yogurt, 50 g cheese
Fish and seafood	≥3 servings/week	1 serving: 130 g fish, 200 g seafood
Lean meat	≥3 servings/week	1 serving: 130 g
Eggs	≥3 servings/week	1 serving: quantity 1
Pulses	≥2 servings/week	1 serving: 150 g
Nuts, preferably chestnuts, walnuts, almonds and hazelnuts	≥4 servings/week	1 serving: 30 g
Fatty meat, cured sausage, margarine, butter	≤4 servings/month	1 serving: 50 g fatty meat or sausage, 12 g margarine or butter
Sweets, pastries, cakes, candies, ice cream	≤4 servings/month	1 serving: 50 g sweets, pastries or cakes, ice cream quantity 1
Sugar-sweetened beverages	≤4 servings/month	1 serving: 200 cc
Moderate and or vigorous physical activity	≥60 min/day	
Each issue scored with one point if it is met		

**Table 2 nutrients-11-00742-t002:** Baseline characteristics and clinical characteristics of the study population.

Measures	Overall (*n* = 791). Mean or *n* (SD or %)
Age (years)	52.3 (12.0)
Gender, Female (*n*, %)	488 (61.7)
Hypertension (*n*, %)	262 (33.1)
Diabetes (*n*, %)	58 (7.3)
Antihypertensive agents (*n*, %)	193 (24.4)
Hypoglycemic agents (*n*, %)	50 (6.3)
Lipid-lowering agents (*n*, %)	155 (19.6)
SBP (mmHg)	123.9 (16.1)
DBP (mmHg)	76.2 (9.8)
Heart rate (bpm)	70.9 (11.5)
Glycated hemoglobin (%)	5.5 (0.6)
Serum glucose (mg/dL)	90.2 (17.5)
Total cholesterol (mg/dL)	204.0 (36.0)
HDL-cholesterol (mg/dL)	58.7 (15.1)
LDL-cholesterol (mg/dL)	124.5 (31.6)
Tryglicerides (mg/dL)	110.4 (65.8)
Fibrinogen (mg/dL)	358.9 (85.1)
BMI (kg/m^2^)	27.8 (4.9)
Waist circumference (cm)	95.1 (13.1)
Hip circumference (cm)	105.0 (9.4)
Waist-height ratio	0.58 (0.78)
Waist-hip ratio	0.91 (0.10)
Adiposity index	27.6 (6.4)
CVR	10.1 (10.4)
Moderate and or vigorous physical activity (min/day)	65.0 (30.9)

Categorical variables are expressed as *n* (%) and continuous variables as mean ± standard deviation (SD). SBP: Systolic blood pressure; DBP: Diastolic blood pressure; HDL: High-density lipoprotein cholesterol; LDL: Low-density lipoprotein cholesterol; BMI: Body mass index; CVR: Cardiovascular risk; cfPWV: Carotid femoral pulse wave velocity.

**Table 3 nutrients-11-00742-t003:** Adherence to the ATLANTIC diet and lifestyle index.

Atlantic Diet Adherence Items	*n* (%)
Bread, cereals, wholegrain cereals, rice, pasta and potatoes (≥6 servings/day)	22 (2.8)
Olive oil (≥3 servings/day)	195 (24.7)
Fresh fruit (≥3 servings/day)	253 (32.0)
Vegetables (≥2 servings/day)	362 (45.8)
Dairy products (≥3 servings/day)	265 (33.5)
Fish and seafood (≥3 servings/week)	716 (90.5)
Lean meat (≥3 servings/week)	513 (64.9)
Eggs (≥3 servings/week)	47 (5.9)
Pulses (≥2 servings/week)	71 (9.0)
Nuts, preferably chestnuts, walnuts, almonds and hazelnuts (≥4 servings/week)	195 (24.7)
Fatty meat, cured sausage, margarine, butter (≤4 servings/month)	71 (9.0)
Sweets, pastries, cakes, candies, ice cream (≤4 servings/month)	144 (18.2)
Sugar-sweetened beverages (≤4 servings/month)	596 (75.3)
Moderate and or vigorous physical activity (≥60 min/day)	409 (51.7)

**Table 4 nutrients-11-00742-t004:** Energy intake and daily nutrients intake by score of ATLANTIC diet and lifestyle index.

	Overall (*n* = 791) Mean or *n* (SD or %)	0–3 Points (*n* = 184) Mean or *n* (SD or %)	4–5 Points (*n* = 308) Mean or *n* (SD or %)	≥6 Points (*n* = 299) Mean or *n* (SD or %)	*p* Value
Energy intake (kcal/day) *#&	2477 (786)	2103 (600)	2447 (759)	2738 (816)	<0.001
Carbohydrate (%) *	42.3 (7.0)	43.4 (6.8)	41.8 (6.9)	42.3 (7.1)	0.046
Protein (%) *#&	17.7 (3.3)	16.7 (2.9)	17.5 (3.4)	18.4 (3.3)	<0.001
Total fat (%)	37.5 (6.4)	37.0 (6.1)	38.0 (6.3)	37.3 (6.7)	0.219
Saturated fat (%) *#	10.9 (2.5)	11.5 (2.5)	11.0 (2.5)	10.4 (2.4)	<0.001
Fiber (g/day) *#&	27 (11)	20 (6)	26 (9)	34 (11)	<0.001
Cholesterol (g/day) *#&	460 (177)	388 (151)	462 (173)	503 (182)	<0.001

Categorical variables are expressed as *n* (%) and continuous variables as mean ± standard deviation. p: statistically significant differences (*p* < 0.05); Post-hoc contrasts (DMS): * Between 0–3 and 4–5 points, # Between 0–3 and ≥6 points, & Between 4–5 and ≥6 points.

**Table 5 nutrients-11-00742-t005:** Cardiovascular risk, mean cardiovascular risk factors and obesity indexes by score of ATLANTIC diet and lifestyle index. ^a^

	Unadjusted Means					Adjusted Means ^b^				
	0–3 Points (*n* = 184) Mean or *n* (SD or %)	4–5 Points (*n* = 308) Mean or *n* (SD or %)	≥6 Points (*n* = 299) Mean or *n* (SD or %)	*p* Trend		0–3 Points (*n* = 184) Mean or *n* (SD or %)	4–5 Points (*n* = 308) Mean or *n* (SD or %)	≥6 Points (*n* = 299) Mean or *n* (SD or %)	*p* trend	
CVR	10.2 (11.1)	9.4 (9.0)	10.6 (11.2)	0.377		12.6 (9.0)	9.7 (8.6)	8.8 (8.9)	<0.001	*#
^C^ SBP (mmHg)	123.5 (18.2)	124.3 (14.8)	123.6 (16.2)	0.829		124.3 (14.4)	124.5 (13.7)	122.9 (14.3)	0.344	
Serum glucose (mg/dL)	90.4 (18.6)	90.0 (17.8)	90.3 (16.5)	0.970		90.9 (14.7)	90.8 (14.0)	89.2 (14.6)	0.326	
Total cholesterol (mg/dL)	199.0 (32.9)	208.4 (37.1)	202.5 (36.0)	0.013	*&	203.1 (36.8)	208.7 (34.6)	199.7 (36.2)	0.007	&
Tryglicerides (mg/dL)	122.7 (70.5)	112.9 (77.4)	100.3 (45.0)	<0.001	#&	122.6 (67.8)	113.3 (63.9)	99.9 (67.0)	0.002	#&
Fibrinogen (mg/dL)	355.9 (90.2)	356.4 (87.1)	363.3 (79.7)	0.572		363.3 (94.5)	356.7 (88.4)	358.3 (93.6)	0.744	
cfPWV (m/s) *n* = 291	7.1 (1.6)	7.1 (1.3)	7.4 (1.3)	0.166		7.4 (5.5)	7.2 (5.2)	7.2 (5.5)	0.050	*#
BMI (kg/m^2^)	28.3 (5.6)	28.2 (5.0)	27.2 (4.2)	0.016	#&	28.6 (5.0)	28.2 (4.7)	27.0 (5.0)	0.001	#&
Waist-height ratio	0.58 (0.08)	0.59 (0.08)	0.58 (0.07)	0.184		0.59 (0.08)	0.59 (0.07)	0.57 (0.08)	0.001	#&
Waist-hip ratio	0.91 (0.10)	0.91 (0.11)	0.90 (0.08)	0.211		0.91 (0.08)	0.91 (0.09)	0.90 (0.09)	0.121	
Adiposity index	27.1 (6.7)	28.0 (6.8)	27.3 (5.8)	0.252		28.2 (6.3)	28.1 (6.0)	26.6 (6.3)	0.004	#&

^a^ Results are reported as mean ± standard deviation. p: statistically significant differences (*p* < 0.05). Post-hoc contrasts (DMS): * Between 0–3 and 4–5 points, # Between 0–3 and ≥6 points, & Between 4–5 and ≥6 points, ^b^ Adjusted means for age, gender and total energy intake by analysis of covariance (ANCOVA). Model for SBP adjusted, also for antihypertensive drugs. Model for serum glucose adjusted, also, for antidiabetic drugs. Model for tryglicerides and total cholesterol adjusted, also, for lipid-lowering drugs. Model for the pulse wave velocity analysis adjusted, also, for antihypertensive drug treatment, blood pressure and smoking status. Model for CVR adjusted only for energy intake. ^c^ SBP: Systolic blood pressure; CVR: Cardiovascular risk; cfPWV: Carotid femoral pulse wave velocity; BMI: Body mass index.

## Data Availability

The datasets used and/or analysed during the current study are available from the corresponding author on reasonable request.

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
