# Peer review of "The Relationship of the Atlantic Diet with Cardiovascular Risk Factors and Markers of Arterial Stiffness in Adults without Cardiovascular Disease"

_nutrients, 2019, doi:10.3390/nu11040742_

Round 1
Reviewer 1 Report
This manuscript by Rodríguez-Martín and colleagues aims to to analyse the relationship of an index measuring compliance with recommendations regarding the Atlantic diet and lifestyle and cardiovascular risk factors, obesity and arterial stiffness markers. In my opinion, this work is well designed and interesting for the scientific community since it is among the few studies investigating this topic.
However, in my opinion, the introduction section does not provide sufficient background and referenfeces. Although the Authors used a novel a priori index to measure the adherence to the Atlantic Diet, the assessment of a posteriori dietary patterns by Principal Component Anlaysis and/or Cluster anaysis is one of the main techniques in this field of research (please consider for example doi: 10.3390/nu10070898 and doi: 10.1016/j.numecd.2018.04.002).
As discussed by the Authors, the cross-sectional design is one of the limitations of this study. However, I recognize that their findings may be useful to propose longitudinal studies to clarify the role of this dietary profile on health.
Methods are adequately described. I only suggest to include the reference period of FFQ (previous week, month or year).
Results are well described and supports the conclusions
In general, I suggest moderate English revision for confusing sentences and typos.
Author Response
Reviewer 1:
This manuscript by Rodríguez-Martín and colleagues aims to analyse the relationship of an index measuring compliance with recommendations regarding the Atlantic diet and lifestyle and cardiovascular risk factors, obesity and arterial stiffness markers. In my opinion, this work is well designed and interesting for the scientific community since it is among the few studies investigating this topic.
Answer:
We wish, first of all, to thank the reviewer for his comments, which we think are very valuable to improve the manuscript.
However, in my opinion, the introduction section does not provide sufficient background and references. Although the Authors used a novel a priori index to measure the adherence to the Atlantic Diet, the assessment of a posteriori dietary patterns by Principal Component Analysis and/or Cluster analysis is one of the main techniques in this field of research (please consider for example doi: 10.3390/nu10070898 and doi: 10.1016/j.numecd.2018.04.002).
Answer:
We share the opinion of the reviewer that the analysis of dietary patterns and their relation to health can be made with different approaches. It is necessary to add the main concepts related to the quality of the diet, adaptation to dietary patterns, analyse its main components or through a cluster analysis, determining, in essence, the actual weight of these components.
We have introduced a paragraph commenting on what the reviewer indicates regarding this type of analysis (line 54):
Some studies complement the analysis of the effect of these dietary patterns on health by analysing the principal component and/or using a cluster analysis. These analyses have been able, for example, to determine that within the concept of a healthy eating pattern, the consumption of a diet rich in cereals, fish, fruit and vegetables is associated with a healthier metabolic profile [1]. They have also allow determining that skipping breakfast or the afternoon snack are risk factors for poor cardiovascular health [2]
As discussed by the Authors, the cross-sectional design is one of the limitations of this study. However, I recognize that their findings may be useful to propose longitudinal studies to clarify the role of this dietary profile on health.
Answer:
Thank you very much for the appreciation. One of our next lines of work will be to study the effect of adherence to this dietary pattern on cardiovascular health through a longitudinal study.
Methods are adequately described. I only suggest to include the reference period of FFQ (previous week, month or year).
Answer:
We have included the reference period of FFQ (previous year) (line 125):
Dietary intake was collected by means of a semiquantitative, self-administered food frequency questionnaire (FFQ), using the previous year as the reference period.
Also, this information is described in the methods (line 128):
After receiving instructions from the study staff, participants indicated the frequency with which each food item was consumed over the last year using a Likert scale of 9 options (never or almost never, 1-3 times a month, once a week, 2-4 times a week, 5-6 times a week, once a day, 2-3 times a day, 4-6 times a day or more than 6 times a day).
Results are well described and supports the conclusions
Answer:
Thank you for your appreciations
In general, I suggest moderate English revision for confusing sentences and typos.
We have sent the manuscript to revise the English to avoid confusing sentences and tipos.
REFERENCES
1. Agodi, A.; Maugeri, A.; Kunzova, S.; Sochor, O.; Bauerova, H.; Kiacova, N.; Barchitta, M.; Vinciguerra, M. Association of Dietary Patterns with Metabolic Syndrome: Results from the Kardiovize Brno 2030 Study. Nutrients 2018, 10, doi:10.3390/nu10070898.
2. Maugeri, A.; Kunzova, S.; Medina-Inojosa, J.R.; Agodi, A.; Barchitta, M.; Homolka, M.; Kiacova, N.; Bauerova, H.; Sochor, O.; Lopez-Jimenez, F., et al. Association between eating time interval and frequency with ideal cardiovascular health: Results from a random sample Czech urban population. Nutrition, metabolism, and cardiovascular diseases : NMCD 2018, 10.1016/j.numecd.2018.04.002, doi:10.1016/j.numecd.2018.04.002.
Reviewer 2 Report
This manuscript summarises the effects of an Atlantic diet score on a wide range of cardiovascular risk factors in a secondary data analysis from the EVIDENT 2 study. It is an interesting paper on the effects of dietary intake on CVD risk factors.
I have a few concerns that could be addressed:
Abstract:
Line 29: included in the analysis, not selected for analysis.
Line 30: without CVD is given as an exclusion here, and this does not match up with what is described in the methods.
Introduction:
There is no acknowledgment of the DASH diet here, and to date, it has provided the strongest evidence for prevention of heart disease. The DASH, Mediterranean and the Atlantic diet have high levels of overlap which need to be acknowledged and the differences between them need to be described clearly.
Materials and methods:
This section needs more work to show how this score was designed, validated and analysed.
Line 89: baseline, not basa.
A brief description of the original study needs to be provided on line 91.
Line 93: Need to describe how the participants became to be part of the original study.
Line 95: Were none of the other measures essential to the analysis? Just the completed FFQ?
A reference is required for line 101 to show that this is a valid way to calculate a sample size. Other dietary scores have been validated using biological markers or nutrients according to the score.
Line 103. What is the DÁgostino scale and why has it been used as the outcome marker for this score? What is the direct link from the AD score to this scale? Also, why has this scale been used for CVD risk when other more validated measures, such as the Framingham exist?
Line 116: What is the name of this FFQ?
Line 117: How was the FFQ validated eg: energy, nutrients, against another method, biomarkers etc? How is the FFQ validated for this kind of diet type?
Line 123: “The results allowed us to calculate…” Why was any kind of analysis done prior to this? This process needs to be included as part of your methods going forward.
Line 124: A lot more detail needs to be added as to HOW the score was put together. What is the validation of using them from other people? Has their work shown them to be successful and if so, why do you need to develop another method?
Why are the score allocated with the weighting? Surely some of the factors here should be weighted more heavily because of their relationship with CVD prevention? Some justification for the scoring with relation to CVD biology needs to be added if the score is to related to CVD prevention, not diet. The lack of weighting may have impacted on results provided here, especially in relation to PA.
Table 1: please give what is considered to be a “serving size” for the foods in this list. How do they relate to country or international dietary recommendations?
Why is physical activity in the list and not smoking status? Given that PA was included as a lifestyle factor, it is not logical that smoking is also not included.
Line 138: Was LDL measure or just calculated using the equation?
Line 165: Why was only one cohort used for arterial stiffness measures and how was this cohort different from the others?
Line 179: Need to describe how cardiovascular risk was calculated to show that it doesn’t overlap with the other confounders mentioned.
Line 199: Where are the p values reported here. Suggest putting in a table showing both unadjusted and adjusted values for the analyses run.
Results:
Line 188: Were any people removed for having unrealistic dietary values? Given that PA measures were available, it seems reasonable to be able to discover those who either under or over-reported their dietary intake.
Table 4: The energy value provided for the cohort seems very high to me. See earlier comment about over and under-reporting.
Discussion
From Line 249: The information provided here would be better placed in the methods, as this relates to how your score was derived.
Line 252-254: Why was it added?
Line 256: Is this reference correct? It links to the validation paper.
The efficacy of the Mediterranean and DASH diets are well established, but what is unclear is how the differences in the Atlantic diet contribute to CVD prevention.
Conclusion: It will be easier to see if this has not been over stated once adjusted and unadjusted values are presented together.
Author Response
We wish, first of all, to thank the reviewer for his comments, which we think are very valuable to improve the manuscript.
Attached is a point-by-point reponse in a pdf file

Round 2
Reviewer 1 Report
All my comments and suggestions have been satisfied.
Author Response
Thank you very much for the review of the article. Your suggestions have been very helpful.
Reviewer 2 Report
There have been many improvements in the paper since the last revision, and the authors are to be congratulated for the fast response.
I still have a few concerns that I believe can be addressed.
L73: Suggest changing SOME to MANY. There are many similarities between these diet types. The only notable difference that relates biologically to CVD is the higher amount of omega three and lean protein from fish and seafood sources.
L77: Check that moderate lean meat isn't a part of the recommended dash diet.
Line 90: Given that your power calculation is based on the CVD risk score, this should be your primary outcome and the others are then secondary.
L111: This paragraph needs to be after the description of how the cardiovascular risk was measured and before the stats section.
L127: Validated for what? Energy and all the macronutrients?
L142: Suggest rewording the sentence to be in third person.
L148: The method for how the PA was collected was included, but where is the data provided?
L170: What measure does the assessment provide? Is it a 5 or 10 year risk assessment? And is this the same one that is used in the power calculation?
L199: This still seems unclear to me. It seems as if the cohort was divided into tertiles before the analysis, based on the calculated score. Would it not have been better to decide on categories aprior?
L206: Given my earlier concerns about energy intake values generated from the FFQ, it is very important to justify inclusion of all energy values or clean the data and re-run the analysis. I believe the second method would be more appropriate. The maximum and minimum energy values seem biologically implausible to me (however, I cannot see their weight status or PA values). I suggest that given you have PA values, that you use something like the Goldberg equation as suitable to remove outliers.
Table 5: CVR should be at the top of the table if this is what is used in the power calculation. (Do you mean CVR, not CR in the footnote?)
Author Response
Attached is a cover letter to explain point-by-point the
details of the revisions in the manuscript and our responses to the
reviewers' comments.
